# Self-Assessed Threshold Temperature for Cold among Poultry Industry Workers in Thailand

**DOI:** 10.3390/ijerph20032067

**Published:** 2023-01-23

**Authors:** Wisanti Laohaudomchok, Wantanee Phanprasit, Pajaree Konthonbut, Chaiyanun Tangtong, Penpatra Sripaiboonkij, Tiina M. Ikäheimo, Jouni J. K. Jaakkola, Simo Näyhä

**Affiliations:** 1Department of Occupational Health and Safety, Faculty of Public Health, Mahidol University, Bangkok 73170, Thailand; 2School of Public Health, Physiotherapy and Sports Science, University College Dublin, D04 V1W8 Dublin, Ireland; 3Research Unit of Population Health, University of Oulu, 90570 Oulu, Finland; 4Department of Community Medicine, University of Tromsø, 9019 Tromsø, Norway

**Keywords:** workplace cold, cold threshold, office worker, sedentary worker, sex, poultry industry, Thailand

## Abstract

The self-assessed threshold temperature for cold in the workplace is not well known. We asked 392 chicken industry workers in Thailand what they regard as the cold threshold (CT) and compared subgroups of workers using linear and quantile regressions by CT sextiles (percentiles P_17_, P_33_, P_50_, P_67_, and P_83_, from warmest to coldest). The variables of interest were sex, office work, and sedentary work, with age, clothing thermal insulation, and alcohol consumption as adjustment factors. The mean CT was 14.6 °C. Office workers had a 6.8 °C higher mean CT than other workers, but the difference ranged from 3.8 °C to 10.0 °C from P_17_ to P_83_. Sedentary workers had a 2.0 °C higher mean CT than others, but the difference increased from 0.5 °C to 3.0 °C through P_17_–P_83_. The mean CT did not differ between sexes, but men had a 1.6–5.0 °C higher CT at P_17_–P_50_ (>20 °C) and a 5.0 °C lower CT at P_83_ (<10 °C). The CT was relatively high at warm (≥10 °C), dry (relative humidity <41%), and drafty (air velocity > 0.35 m/s) worksites. We conclude that office, sedentary, and female workers and those working at warm, dry, and draughty sites are sensitive to the coldest temperatures, whereas male workers are sensitive even to moderate temperatures.

## 1. Introduction

Poultry industry workers, who often work at temperatures down to −20 °C, frequently suffer from cold-related harm. In Thailand, for example, most of them report poor work performance, or musculoskeletal or respiratory symptoms [1,2,3,4,5]. Studies conducted elsewhere have shown a wide variety of cold-related respiratory, cardiovascular, and musculoskeletal symptoms, worsening of chronic medical conditions [6,7,8,9,10] and poor work ability [11] among them. Cold-related adversities are not limited to symptoms since cold symptoms predict actual disease episodes and deaths due to them [12]. Populations living in a tropical climate are especially sensitive to low temperatures, likely due to a lack of adaptation to the cold. In Bangkok, Thailand, all-cause mortality among the population starts to increase when environmental temperature declines below 29 °C, while the threshold is 15–20 °C in a temperate climate [13]. At workplaces, the harmful effects of the cold are largely preventable by appropriate clothing, regulating the time spent in the cold, and health education [14]. It is therefore useful to know not only the prevalence of cold adversities but also the threshold temperature at which the workers start to perceive cold and then likely take protective actions. It is also important to identify any sensitive subgroups of workers who perceive higher temperatures than others as cold. This would be an aid for targeting preventive actions and providing a rational basis for the regulation of indoor temperature, which also has obvious implications regarding energy consumption and air pollution.

Thailand is a major producer of chicken meat and its products, worldwide. In 2020–2021, the country was ranked as the fourth largest exporter after Brazil, the USA, and the EU [15,16]. There are 537 chicken slaughter factories (with the capability of slaughtering 5.03 million chickens per day) and 69 chicken product processing plants throughout the country. A total of 2.5 million tons of chicken meat was produced, and 0.97 million tons were exported in 2021 [17]. The export value of chicken meat and its products was THB 102,527 million (USD 3.206 million) in 2021 [18]. Among the total labour force of 38.7 million, 5.9 million work in manufacturing including the chicken meat industry [19]. Thus, any positive effects from the prevention of cold harm would affect a large number of workers and would produce marked improvements in terms of cold-related health burden and productivity.

In our previous study, we asked poultry industry workers in Thailand what temperature they perceived as cold at the workplace [1]. The most common answer was 20 °C, especially office workers and highly-educated workers reporting this temperature as cold. Except for this preliminary observation, we are not aware of any in-depth analyses of this topic. The present study determines the self-assessed cold temperature (CT) among poultry industry workers with breakdowns according to sex, age, job category, education, physical work strain, thermal insulation of clothing, obesity, alcohol consumption, and temperature, relative humidity, and air velocity at the worksite. In particular, we aimed to quantitatively estimate the differences in CT in the above subgroups of workers compared with other workers, allowing for relevant confounding factors and also looking for variations of group-wise differences at different levels of CT.

## 2. Materials and Methods

### 2.1. Study Population

We used a database of poultry industry workers described previously in detail [1]. To determine the occurrence of cold-related harm, participants were recruited from four chicken meat factories in northeastern Thailand with 288, 5054, 500, and 7250 workers, respectively, totalling 13,092 men and women. The data were collected during July–December 2017, when daytime outdoor temperatures ranged from 28 °C to 34 °C. Based on the power calculations, 422 employees in the factories concerned (59, 145, 70, and 148, respectively) were selected using convenience sampling. The number of workers in each factory was dictated by their availability during regular working hours and the work schedule of the study team. Of the 422 workers recruited, 392 employees with information on relevant personal and workplace factors were included in this analysis. Worksite temperature (Ta), relative humidity (RH), and air velocity (AV) were recorded in a subsample consisting of 284 workers in cold storage and manufacturing halls and in offices in which workers were willing to participate. The subsample was used as an additional data source to describe workplace physical conditions and assess the roles of Ta, RH, and AV in the results. The work in the factories consisted of chicken meat cutting, processing, storing, packing, and paperwork, and took place in cold storage, manufacturing halls, and office premises. Manufacturing workers cut and shape chicken meat and package it, forklift drivers transport the packages to cold storages and shipping yards, storage workers move and lift the packages in cold storages, and office workers normally do paperwork in offices.

### 2.2. Interview

Information on each participant’s personal details, work-related factors, living habits, and cold-related symptoms was obtained through a structured interview conducted by trained interviewers (Figure A1). The self-assessed cold threshold was determined by the question, “What temperature do you regard as cold? (°C)” which was used as a measure of cold sensitivity. This question has been used by the FINRISK 2002 study team comprising specialists in occupational health, epidemiology, physical medicine, and physiology [20]. The respondents were asked to indicate their job category (manufacturing worker, cold storage worker, forklift driver, or office staff), which was dichotomised as office work vs. other work. Physical strain at work was elicited by asking whether the work was sedentary, other light work, medium heavy work, or heavy work, and was dichotomised as sedentary vs. other work. Further questions were asked about the highest attained educational level (high: university or vocational school; low: high school, middle school, primary school, or less). We also asked about body weight and height, smoking status (current smoker; other: previously smoked, never smoked), alcohol consumption (weekly; other: occasionally, monthly, never), and leisure time physical exercise (times/week). The participants were asked how many times a day they moved between cold and warm sites (4+ times/less often) and how many hours a day they spent at temperatures below 0 °C. The interviewees were also asked to indicate the clothing items they used at work (28 separate items) [21]. The basic thermal insulation of the clothing ensemble (I_cl_) was calculated as I_cl_ = 0.161 + 0.835 ∑ I_clu_, where I_clu_ denotes the clo value for each clothing item.

### 2.3. Measurements

Air temperature (Ta) and relative humidity (RH) were measured using a 303 C thermo-hygrometer (Shenzhen Graigar Technology, Shenzhen, China), and air velocity (AV) was measured using a VelociCalc^®^ 9545 (TSI Incorporated, Shoreview, MN, USA). The technical details of the instruments used are in Table A1. The measurements were conducted at 37 points in the participant’s regular working area, which was determined by the question, ‘Where do you work most of the time?’. Twenty sites were in cold storage, thirteen in manufacturing halls, and four in offices. In areas where the temperature and air velocity varied, the minimum and maximum values were recorded and expressed as average values. The values recorded for each working area were linked to each participant in the study.

### 2.4. Data Analysis

The self-assessed CTs (°C) were first described in terms of means, medians, SDs, and ranges; their group-wise differences were tested using one-way analysis of variance and linear regression coefficients. The subgroup differences were further elaborated graphically by presenting the cumulative distribution of CT in each subgroup, with values accumulating from warm to cold. These suggested that group-wise differences in CT may be different at various locations of its distribution. Therefore, quantile regression [22] was used to quantitatively determine the association between CT and explanatory factors at various levels of CT. This analysis uses predefined quantiles of CT as response variates and allows the association to vary at different levels of CT. The quantiles of CT were determined by assigning 1/6 of the observations to each class (corresponding to percentiles P_17_, P_33_, P_50_, P_67_, and P_83,_ from warmest to coldest), and quantile regressions were conducted using these percentiles as the response variates. Sextiles were used because they described the response pattern in more detail than lower quantiles (tertiles to quintiles) and were less erratic than higher quantiles (septiles to deciles). The explanatory factors of main interest were sex, job category, and physical work strain, whereas age, thermal insulation of clothing, and alcohol consumption were included as adjustment factors. Additional analyses allowed for workplace Ta, RH and AV, but no attempt was made to quantify their independent effects. Education, smoking, leisure-time exercise, and moving between cold and warm sites were also considered as adjustment factors, but they had only marginal effects on the parameters of interest and were omitted from the final analyses. The models were fitted using a variant of the Barrodale and Roberts algorithm [23]. The results were expressed as regression coefficients indicating the differences in CT in each group vs. the reference group, and the confidence intervals were obtained using the bootstrap method with 5000 replications. The coefficients were calculated using the rq function available in R software, release 4.1.2 (https://CRAN.R-project.org/, accessed on 21 January 2023). The results were also presented as marginal means for each variable of interest, involving the calculation of averages of all model-estimated CTs for the outcome [24,25]. Each marginal estimate represented the CT adjusted for all other factors in the model.

## 3. Results

### 3.1. Characteristics of the Subjects and Variations between Subgroups

#### 3.1.1. Personal Characteristics

Table 1 summarises the participants’ characteristics. There were 392 workers, of whom 186 (47%) were men. The mean age was 33.1 years (SD 9.9; range 18–57), and 219 (56%) were aged ≥ 30 years. Altogether, 128 participants (33%) were office staff (83% were women), and the remaining 264 (67%) were manufacturing or cold storage workers or forklift drivers. Altogether, 43% of all participants and 91% of office staff were highly educated. A total of 173 participants (44%) engaged in heavy physical work, 219 (56%) engaged in light physical work, and 132 (34%) engaged in sedentary work. The majority of office staff (61%) were sedentary, but sedentary work was not uncommon (21%) in other job categories.

The body mass index (BMI) averaged 25.2 kg/m^2^ (SD 4.8, range 15.6–44.1), and 147 participants (38%) were classified as obese (BMI ≥ 25.0 kg/m^2^). Ninety-six participants (24%) were smokers, 46 (12%) consumed alcohol on a weekly basis, and 182 (46.4%) exercised at least once a week during leisure time. Twenty-six participants (6.7%) reported diagnosed cardiovascular disease, 20 (5.1%) reported a respiratory disease, and 25 (6.4%) reported a back, joint, skin, or mental condition or diabetes mellitus.

#### 3.1.2. Work-Related Factors

The mean daily working time was 8.7 h per day (men 8.9 h/day, women 8.4 h/day), with 54% working > 8 h/day. Office workers did slightly fewer hours (8.3 h/day) than other workers (8.9 h/day) and fewer of them worked > 8 h/day (25% vs. 68%, respectively). Sedentary and non-sedentary workers showed no major difference in average working time (8.6 vs. 8.8 h/day, respectively) nor percentage working > 8 h/day (46 vs. 58%, respectively).

The mean thermal insulation of clothing was 1.15 clo (SD 0.30, range 0.35–2.21), and it was lower in women (1.08 clo) than in men (1.20 clo), lower among office workers (0.85 clo) than others (1.27 clo), and lower among sedentary (1.03 clo) than non-sedentary workers (1.19 clo). The mean clo was similarly low among office workers independent of whether the work was sedentary or non-sedentary (0.82 vs. 0.90, respectively), and it was similarly high among sedentary and non-sedentary non-office workers (1.32 vs. 1.26, respectively).

Altogether, 308 subjects (78.8%) moved daily between cold and warmer sites at least four times a day, with 78.5% and 79.0% for men and women, and 70.3% and 82.9% for office workers and other workers, respectively. The participants spent an average of 1.59 h per day (range 0–10) at temperatures <0 °C, with men and women spending 2.92 h/day and 0.40 h/day, respectively, office workers and other workers spending 0.19 h/day and 2.87 h/day, respectively, and sedentary and non-sedentary workers spending 0.81 h/day and 1.99 h /day, respectively. All participants reported that they had air conditioning in the workplace.

#### 3.1.3. Workplace Physical Conditions

Measurements of Ta, RH, and AV for a subsample of 284 workers are described in Table A2. The mean Ta was 3.8 °C (SD 11.4 °C, range −21.5–23.0) and, according to ISO standard [26], it was classified as cold (<10 °C) for 64.4% of the workers. RH averaged 46.8% (SD 12.9, range 27.0–72.0). AV had a skewed distribution with a long right-hand tail, with a median of 0.38 m/s and a range of 0.01–3.00 m/s.

Women worked at warmer temperatures than men (mean Ta 10.4 °C vs. −1.1 °C, respectively), and their worksites were more humid (RH 55.0% vs. 40.8%, respectively) while AV was similar for women and men (median 0.36 m/s vs. 0.40 m/s, respectively). Office workers experienced higher temperatures than other workers (mean 22.3 °C vs. 1.7 °C, respectively), and their worksites were more humid (RH 63.8% vs. 44.9%, respectively) and drafty (AV 0.44 m/s vs. 0.34 m/s, respectively). Sedentary and non-sedentary workers showed little difference regarding worksite Ta (2.8 °C vs. 4.1 °C, respectively) and RH (48.2% vs. 46.4%, respectively), but sedentary workers experienced more drafty sites than others (AV 0.44 m/s vs. 0.30 m/s, respectively). Highly educated workers had higher worksite temperatures than low-educated workers (mean Ta 10.2 °C vs. 1.5 °C, respectively), and they worked at more humid (RH 52.5% vs. 44.7%, respectively) and drafty sites (AV 0.44 m/s vs. 0.34 m/s, respectively).

### 3.2. Self-Assessed Cold Threshold

#### 3.2.1. Distribution of CT

The mean CT was 14.6 °C (SD 10.0; range −28.0–30.0), but the distribution was skewed to the warmer side, with a median of 18.0 °C, and 90% of the values located within the range −4.0–27.0 °C (Figure 1). Relatively high temperatures were assessed as cold: 25 °C, for example, by 31 participants (8%) and 20 °C by 179 (46%). Temperatures of ≥10 °C—those equal to or warmer than the ISO-defined limit for workplace cold—were regarded as cold by 326 participants (83%).

#### 3.2.2. Subgroup Differences of the Mean CT

Table 1 also compares the mean and median CT values for personal and workplace characteristics. The mean CT was 2.8 °C lower for men than women (95% CI −4.4–1.1), 6.9 °C higher for office workers than other workers (95% CI 5.2–8.6), 3.8 °C higher (95% CI 2.1–5.4) for highly educated than low-educated workers, and higher by 4.3 °C (95% CI 2.3–6.4) for sedentary than non-sedentary workers. Low thermal insulation of clothing (clo < 1.15) was associated with a 3.1 °C (95% CI 2.1–5.4) higher CT than higher insulation. Smokers had a marginally lower CT than non-smokers, whereas age, obesity, and alcohol consumption were not associated with CT. Participants who exercised at least once a week during their leisure time had 1.8 °C lower CT (95% CI −3.8–0.2) than others.

#### 3.2.3. Cumulative Distributions of CI by Subgroups

Subgroup differences larger than those between the mean CT values are shown in Figure 2 at various locations of the CT distribution. Men reported 10–20 °C lower CTs than women, but only at the cold end of the distribution (CT < 10 °C or at P_80_–P_90_), while no sex differences were seen at CT values ≥ 10 °C. Office workers reported CTs of at least 10 °C higher than other workers, mainly at colder values of CT, although differences existed throughout the distribution. The CT was as much as 10–15 °C higher among the highly educated than the low-educated workers, but mainly at percentiles P_90_ and higher.

Sedentary workers reported 5–15 °C higher CTs than non-sedentary workers at percentiles P_40_ and above (Figure 2). Participants with low thermal insulation of clothing (<1.15 clo) had up to 5 °C higher CT than those with more insulation, mainly around the central locations of the CT distribution. The CT was 10 °C lower among smokers than non-smokers, but only around the 80th percentile, while no marked differences depending on age, obesity, or alcohol consumption were seen at any percentiles of CT.

Figure 2 further shows that participants who worked at warmer sites (≥ 10 °C) had CTs at least 10 °C higher than those working at colder sites but only at the cold end of the CT distribution (P_80_ and higher), while little differences were seen at lower percentiles. Participants at dry working sites (RH < 41%) reported as cold 5–8 °C higher temperatures than those working at more humid sites but mainly at central values of the CT distribution. Workers at drafty working sites (air velocity > 0.38 m/s) assessed as cold 5–20 °C higher temperatures than those at less drafty sites but only at P_40_ or higher.

### 3.3. Regression of Self-Assessed Threshold for Cold

Table 2 shows the regression of CT on sex, age, job category, physical work strain, thermal insulation of clothing, and alcohol consumption after adjusting for personal and workplace factors. The regression coefficients indicate the crude and adjusted differences in CT (°C) in the classes of each explanatory factor compared to a reference class. Figure 3 shows the main findings in the form of model-based adjusted CTs and their differences.

The crude sex differences in Table 2 repeat the empirical observation in Figure 2 at selected percentiles, with a 9.0 °C lower CT in men than women in the coldest percentile, P_83_. However, the adjustment for personal and work-related factors reduced this deficit at P_83_ to 5.0 °C and turned the non-existent crude differences at P_17_–P_50_ to male excesses of 1.6–5.0 °C, with no marked changes after further adjustments for workplace Ta, RH, and AV. Figure 3 shows the model-adjusted CTs, illustrating the higher CT in men than in women at P_17_–P_50_, the contrary difference at P83, and the resulting crossover of male and female CTs.

While the crude mean CT in Table 1 was higher by 6.9 °C (95% CI 5.2–8.5) for office workers than others, the difference persisted after adjustments for personal and workplace factors in Table 2. However, in quantile regression, the difference grew from the lowest to the highest percentile of CT, the crude differences from 4.0 °C to 10.0 °C, those adjusted for personal and work-related factors from 3.8 °C to 10.0 °C, and those additionally adjusted for Ta, RH, and AV from 7.2 °C to 15.0 °C. The steep growth of the coefficients across percentiles was largely due to the steeper accumulation of cases from P_17_ to P_33_ among the office workers (Figure 3).

The mean CT was 4.3 °C higher (95% CI 2.6, 6.1) among sedentary than non-sedentary workers, a difference that reduced to 2.0 °C (95% CI 0.1–3.8) after adjusting for personal and workplace factors, and almost vanished when further adjusted for Ta, RH, and V (Table 2). However, the crude quantile regression coefficients grew by percentiles, from 2.0 °C to 7.0 °C from the lowest to the highest percentile, with somewhat smaller coefficients (0.5 °C to 3.0 °C) after adjustments for personal and workplace factors and one significant coefficient (2.0 °C, at P_17_) remaining after all adjustments. The latter trends were based on a slightly steeper accumulation of CTs across percentiles among sedentary workers than among non-sedentary workers (Figure 3).

Low thermal insulation of clothing (clo < 1.15) was associated with a 3.1 C (CI 1.2–5.1) higher mean CT compared with higher insulation (clo ≥ 1.15), with inconsistent variations of coefficients at various percentiles; however, this was reduced to insignificance after all adjustments (Table 2). The mean CT was unaffected by alcohol consumption, but CT adjusted for personal factors was elevated by 5.0 °C (CI 1.5–8.5) at P_67_ and also elevated at P_83_; however, almost no significant coefficients remained after adjustments for Ta, RH, and AV.

### 3.4. Self-Assessed Cold Threshold in Relation to Worksite Physical Conditions

Working at warmer sites (Ta ≥ 10 °C) was associated with a 1.8 °C higher mean CT (CI −0.2–3.9) compared with working at colder sites, but a 7.0 °C higher CT (CI 2.0–12.0) was seen at P_83_ (Table A3). A dry worksite (RH < 41%) was associated with a mean CT 3.4 °C higher (CI 1.4–5.3) than that at more humid sites but 5.0–6.0 °C higher at the colder percentiles of CT. A drafty worksite (air velocity > 0.38 m/s) was associated with a 3.6 °C (CI 1.6–5.5) higher mean CT compared with a less drafty worksite, with 0.0–6.0 °C higher CTs at various percentiles.

## 4. Discussion

### 4.1. Summary of Findings

There was wide variation in the temperatures that the participants regarded as cold, from −28 °C to 30 °C, with 90% of values located within the range of −4.0 °C to 27 °C and one half at 18.0 °C or higher. Temperatures warmer than 10 °C, the ISO-defined temperature for workplace cold [26], were actually regarded as cold by 83% of the participants. The wide range of variation in CT was also expressed as marked differences between the subgroups of workers, indicating a mix of vulnerabilities. Moreover, the CT distribution was skewed to the warmer side, with different degrees of skewness across the subgroups. This questions the customary use of any single values to represent CT, either in the form of means, or low limits of the “normal”, comfortable, or neutral range [27,28,29]. The benefit of comparing the entire distributions of CT was illustrated here in locating subgroup differences: (1) there was no sex difference in the adjusted mean CTs but significant differences at high and low CTs, (2) a markedly high CT was found for office workers but mainly at colder CT values, and (3) CT was higher for sedentary than non-sedentary workers but more so at colder than warmer CTs. Our study adds to what is previously known about cold sensitivity in this industry and, in particular, quantitatively shows the group-wise differences in CT at various levels. Identifying sensitive groups of workers is useful because cold exposure adversely affects physical and mental performance, and work capacity, and it increases the risk of cold injuries, accidents, and cardiac and respiratory events [29]. Obviously, workers with higher CT are more willing to take protective actions and could thereby reduce the burden imposed by the cold in this industry.

### 4.2. Self-Assessed Threshold for Cold Temperature

Although the mean and median CTs observed, i.e., 14.5 °C and 18.0 °C, respectively, were not directly based on actual cold exposure, but rather on the worker’s image of his/her past thermal sensations of cold, they are close to or slightly below various estimates defined as cool or cold, for example: 18 °C in Taiwan [30], 18 °C in the United Kingdom [9], 17 °C in Harbin, China [31], and 14 °C in Iran [32]. Temperatures of 24–26 °C are mostly regarded as neutral in hot climates but this is 15–20 °C in colder climates [33], reflecting adaptation to the climate of residence. The critical limit at which heat production is initiated to maintain body heat balance in a lightly clothed individual is approximately 22–29 °C [34,35]. In accordance with this, 26 °C has been suggested as the recommended neutral indoor temperature in Thailand [36]. Our result is also consistent with relevant bodily responses to the cold, since with declining ambient temperature, blood pressure can increase significantly at 18.0 °C [37]—a temperature cold enough to also produce haematological changes that predispose one to vascular thrombosis and subsequent cardiovascular events [9,37].

### 4.3. Vulnerable Groups

Our main finding was the 6.8 °C higher CT among office workers compared with that among other workers. Our previous study found that a higher proportion of office workers regarded temperatures > 20 °C as cold, and more office workers suffered from cold-related cardiac symptoms and musculoskeletal pain, as well as poorer concentration and motivation [1,2]. In addition, the number of individual cold-related symptoms is relatively high among office workers [1]. In the present analysis, not only was the mean CT high among office workers, but CT was particularly high in the colder part of its distribution. Thus, among all participants who assessed temperatures lower than approximately 15 °C as cold, the adjusted CT was higher by 10–15 °C for office workers than others, while the difference was 4–7 °C for those who assessed temperatures > 15 °C as cold.

The thermal insulation of clothing as low as 0.85 clo among office workers provides an obvious explanation for their relatively high CT. Clo values lower than the median 1.15 were associated with a 3.1 °C higher CT in the crude analysis, although this factor did not appear as an independent factor, likely due to its effect mixing with other factors. Thermal insulation was allowed in the analysis; however, office workers frequently suffer from cold symptoms in the cheeks, ears, and elbows, which often remain unprotected [2]. Other explanations could be the worsening of cognitive functions that reportedly occur at temperatures <14 °C [32]. Additionally, prolonged use of a computer mouse at office temperatures may decline hand skin temperatures by 5 °C [38] and thereby affect reporting of CT. Additional factors among office workers include high relative humidity and air velocity, and possibly air conditioning that is too effective [39]. These factors may explain why office workers regard relatively high temperatures as being cold. The reason why the difference in CT increased from warmer to colder percentiles is not immediately clear. Possibly, occasional visits to sites 10–40 °C colder than the regular worksite have affected the worker’s image of cold and made a greater difference the colder the site visited.

Much of the reason why office workers have a high CT relates to the question of why they do not wrap up adequately. One could assume that they have mistaken beliefs of office temperatures of around 20 °C not being harmful because these are much higher than the standard definition for workplace cold of 10 °C [26]. In addition, some contextual effects not captured by individual-level data could play a role, such as dress codes applied at workplaces or fashion trends furtively adopted within occupational subgroups, sometimes referred to as occupational [40] or academic “tribes” [41]. The observed association of a high CT with office work may have been mixed with the effects of higher education, which is a factor in cold-related symptoms [3].

While 61% of the office staff were sedentary, sedentariness was also associated with CT independent of office work. We have previously seen that sedentariness among chicken industry workers is associated with cold-related pain in the knees, lower back, and thighs [2]. A higher CT among sedentary than non-sedentary workers was not unexpected considering their low level of physical work strain and the decline of microcirculation due to permanent muscular contraction to maintain a stable posture [38]. However, no previous study has quantified this difference in terms of CT or at various locations of its distribution. Here, sedentariness was associated with a 7.0 °C higher CT at its coldest sextile, which was partly explained by personal and work-related factors and finally diminished to a difference of 1.5–2.0 °C at the warmest percentiles when adjusted for all factors considered. We conclude that sedentariness is a predisposing factor for high CT, among both office and non-office workers. Exposure to cold temperatures is known to increase the risk of blood clotting, especially among sedentary workers wearing insufficient clothing [9].

According to previous reports, women are more sensitive to cold than men [42,43,44,45,46]. In the food industry, female workers suffer more from cold-related harm than male workers [5,11,45]. With declining temperatures, women become aware of thermal discomfort before males [46]. Suggested explanations include women’s larger surface area to mass ratio, favouring heat loss to the environment, greater propensity to express cold discomfort [47], and health-related issues in general [48]. Our previous analyses revealed more cold-related cardiac and circulatory symptoms in women than in men [1]. Here, the adjusted linear regression showed no overall sex difference in CT, but men had a 1.6–5.0 °C higher adjusted CT at its warmer percentiles and a 5.0 °C lower CT at the coldest percentile.

This implies that, among all workers who regard temperatures warmer than approximately 15 °C as cold, CT is up to 5.0 °C higher in men, while among those assessing temperatures <15 °C as cold, the threshold is 5.0 °C lower in men. This may be interpreted in terms of men possibly using adequate protection only at temperatures they consider very cold, but fail to do so at more moderate temperatures, which women, in turn, recognise as cold enough to require protection. Men are known to accept very cold temperatures more likely than women, while fewer sex differences exist regarding moderate cold [43]. Our results suggest that male workers may be sensitive to moderate cold, given the protective clothing they use, while they would tolerate severe cold better than female workers. Thus, the female disadvantage reported by the studies cited above may be restricted to severe cold, while in moderate cold, women do better than men, probably because of more appropriate protection. Therefore, not only should women be advised to wrap up better in the cold, but men also need to be reminded to do so, even if the worksite temperature is only slightly cold.

### 4.4. Other Factors Affecting CT

It has been reported that older people prefer slightly higher temperatures than younger people because of decreasing metabolic rates [28] and impaired thermoregulation [49] with age. However, we have previously observed that older food industry workers have fewer cold-related symptoms than younger workers [1]. This is in line with our present observation that CT declined by 2.0–3.0 °C every 10 years of age at the coldest percentiles of CT. This could be ascribed to age-related selection, and older individuals with higher stamina staying at work.

As obese people are less sensitive to cold than others owing to their thicker subcutaneous fat layer and greater lean body mass, which increases heat production [50], we expected to see a relatively low CT among obese individuals. Here, the crude mean CT was lower by 0.9 °C among obese than normal-weight workers, but this estimate was inaccurate and did not persist in the adjusted analysis. However, this finding is in line with that of Karyono [28], who noted a 1.0 °C lower neutral temperature among the fat compared with the thin.

Chicken industry workers who consume alcohol weekly or more often have an increased occurrence of cold-related performance problems [1]. We observed 4.0–5.0 °C higher CTs at their coldest percentiles, suggesting higher cold sensitivity among weekly consumers. The immediate effects of alcohol would not provide an explanation, since alcohol ingestion decreases the feeling of cold and related discomfort [51] and delays the start of shivering [52]. Because alcohol is likely only consumed during leisure time, its impacts on thermal responses, such as sweating, shivering, and cold flashes following the day of alcohol consumption [53] may have resulted in these participants reporting elevated CTs at colder values.

Smokers had a 1.8 °C lower mean CT than non-smokers, although this was not a significant factor in the multivariable analysis. This finding was unexpected because smoking reduces cutaneous circulation [54] and microvascular function [55], which could manifest as higher cold sensitivity. Although the effects of smoking on thermal sensations are difficult to assess, other studies have reported a low prevalence of cold-related circulation disturbances among smokers [1] and a lower prevalence of cold pain in the feet and ears [2], and hands, wrists, feet, and ankles among smokers [4]. We believe that the effects of smoking on cold adversities in occupational settings require further research.

Since repeated daily exposure to cold leads to blunting of physiological responses and adjustments to thermal stimuli [56,57], cold-exposed workers can be expected to have a lower CT than less-exposed workers. This is shown here as a relatively low CT in workers at colder sites and higher CTs at warmer sites. We interpret this as an indication of adaptation to the worksite temperature. We also observed an association between low relative humidity RH (<41%) and elevated CT. This is consistent with the knowledge that lowered humidity in the workplace may cause a decrease in skin temperature and thereby a feeling of cold independent of ambient temperature [58]. In addition, high air velocity was associated with a higher CT.

### 4.5. Strengths and Limitations

The main strength of our study is that we focused on variations in the self-assessed cold between subgroups of workers. The need for subgroup analyses to identify vulnerable groups has been recognised [28,33], but there is no comprehensive study on subgroup differences in CT in occupational settings, even though it should be useful in targeting preventive actions. Another strength is that we analysed subgroup differences not only by comparing the mean CTs but also by looking at various locations of its distribution. Without doing so, we would have missed the anomalous sex difference in CT that was directionally different for cold and warmer CTs; additionally, we would have underestimated the markedly high CTs among office workers at their coldest values, and we would have missed some variation of CT depending on sedentariness, alcohol consumption, and workplace physical conditions. We also adjusted for a reasonable number of confounders, which has not always been performed in studies in this field. Our results are based on a well-examined database that provides a reference for cold-related adversities in this industry [1,2,3,14].

The validity of the question regarding the measurement of CT can only be assessed in terms of face validity because the answers cannot be compared with any gold standard. The question was clear and easily understandable, although we did not specify the meaning of cold: a general feeling of cold, feeling cold while breathing, or touching cold surfaces. The question relates to a true experience of cold at the participant’s regular worksite or other sites in his/her workplace and is likely to detect group-wise differences in cold sensitivity. However, establishing the content validity and predictive value of subjectively assessed CT requires further study, preferably in terms of prospective studies.

Convenience sampling was used, which was dictated by the availability of workers during their regular working hours and the working schedules of the study team. This may have affected the results in an unknown way, but any major bias in the group-wise differences is unlikely because we adjusted for relevant confounders. One limitation is that the independent effects of all explanatory factors may not have been identified with certainty because of collinearity. Several medical conditions and medications may further affect cold sensitivity [59], but we were unable to assess their effects due to the small number of sick individuals. The female participants were not asked about their menstrual period, which could affect thermal sensations. Because the study was conducted in a tropical climate, the results cannot be generalised to other climates.

### 4.6. Conclusions

Asking food industry workers about their subjective cold threshold is useful not only because it reflects the actual experience of cold but also because people are likely to take protective action at temperatures which they regard as cold. Our study provides an obvious case for preventive action, especially since we identified subgroups of workers who perceive markedly higher temperatures as cold compared to that of others. Occupational health care should advise office staff and sedentary workers to wear thicker clothing and more clothing layers, even if they do not regard them as appropriate or necessary. Male workers should be informed that even moderate cold requires protection, and female workers who regard only very low temperatures as cold need more clothing. The recommended values for workplace temperatures should preferably be customised based on their entire distribution, and not only on any single value or range. Additional measures to be targeted to office and sedentary workers and to men and women include drinking hot beverages, keeping hydrated, avoiding unnecessary stays in the cold, and setting appropriate air conditioning. The regulation of indoor temperatures, allowing for different perceptions of cold among various categories of workers, should be considered. Research-based recommendations and standards for cold work that do not currently exist in Thailand should be established, considering variations between subgroups of workers. Further research is needed to explain the reasons for variations in cold perception. This information is especially relevant in view of ongoing climate change, which will affect this region the most [60] and will increase the contrast between tropical outdoor heat and cold within cold workplaces.

## Figures and Tables

**Figure 1 ijerph-20-02067-f001:**
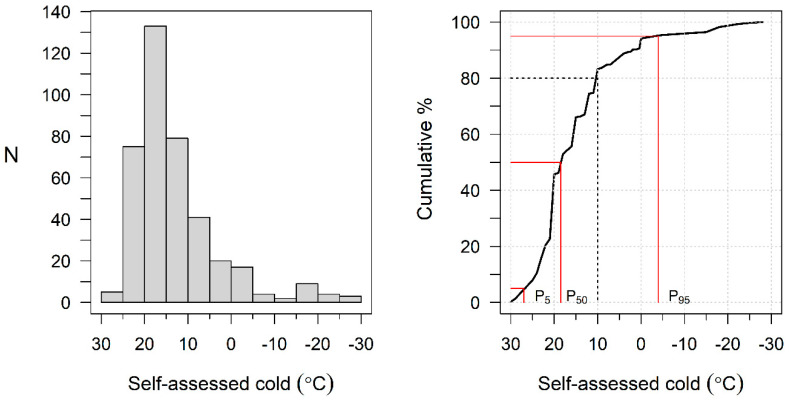
Histogram and cumulative distribution of self-assessed cold temperature. Dotted line indicates the ISO-defined threshold for workplace cold (10 °C).

**Figure 2 ijerph-20-02067-f002:**
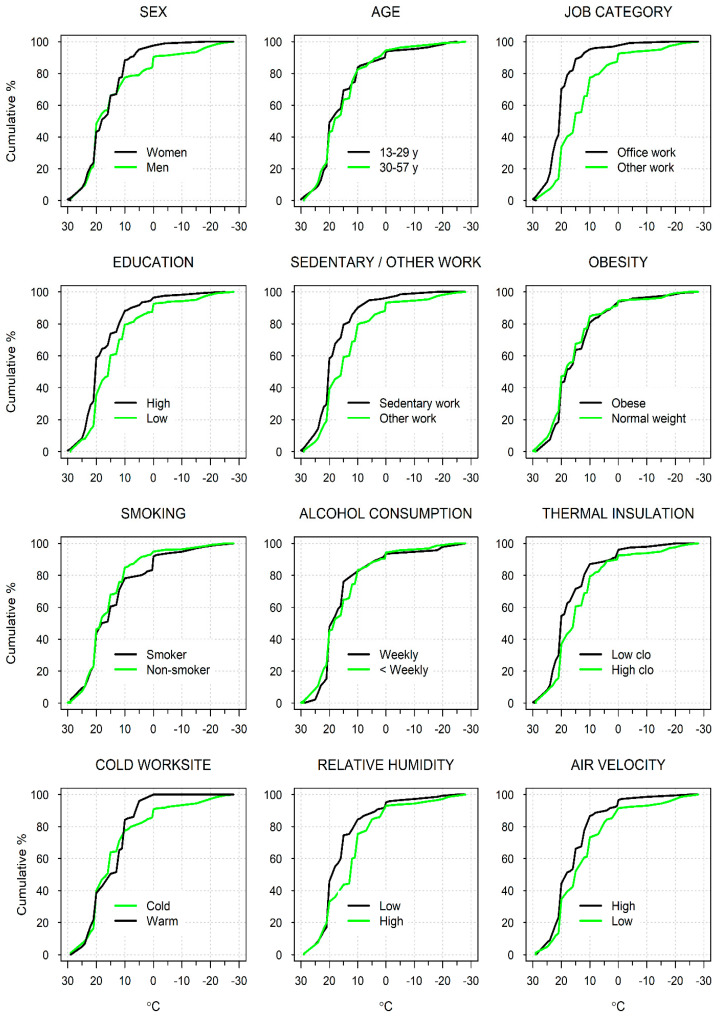
Cumulative distributions of self-assessed threshold for cold temperature (°C) according to personal and workplace characteristics and physical conditions at workplace.

**Figure 3 ijerph-20-02067-f003:**
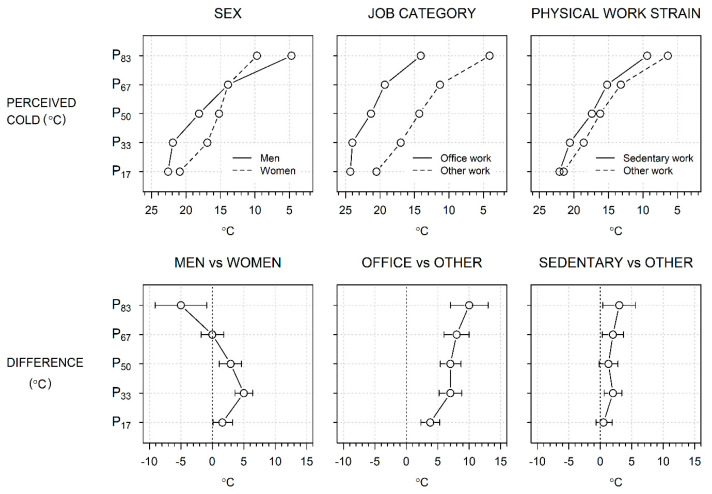
Model-based adjusted threshold for cold (°C) by percentiles (P_17_: warmest, P_83_: coldest), broken down by sex, job category, and physical strain at work (upper row) and their groupwise differences (lower row).

**Table 1 ijerph-20-02067-t001:** Self-assessed threshold for cold temperature according to personal and workplace factors.

Factor	Mean (SD) (°C)	Difference (°C)	Median (°C)	Range (°C)	N (%)
Sex					
Men	13.2 (12.3)	−2.8 (−4.4, 1.1)	18.0	−18, 30	186 (47)
Women	16.0 (7.3)	Ref.	18.0	−28, 29	206 (53)
Age					
30–57 yr	14.6 (9.8)	−0.1 (−1.8, 1.6)	18.0	−28, 29	219 (56)
18–29 yr	14.7 (10.4)	Ref.	18.0	−25, 30	173 (44)
Job category					
Office work	19.3 (6.4)	6.9 (5.2, 8.6)	20.0	−18, 30	128 (33)
Other work	12.4 (10.7)	Ref.	15.0	−28, 29	264 (67)
Education					
High	16.8 (8.5)	3.8 (2.1, 5.4)	20.0	−25, 25	168 (43)
Low	13.0 (10.8)	Ref.	15.0	−28, 29	224 (57)
Physical work strain					
Sedentary work	17.5 (7.7)	4.3 (2.3, 6.4)	20.0	−18, 30	132 (34)
Other work	13.2 (10.8)	Ref.	15.0	−28, 29	260 (66)
Thermal insulation of clothing					
Low (clo below median, 1.15)	16.2 (8.8)	3.1 (1.2, 5.1)	20.0	−20, 30	194 (50)
High (clo at least median, 1.15)	13.1 (10.9)	Ref.	15.0	−28, 29	198 (50)
Obesity					
Obese (BMI ≥ 25.0 kg/m^2^)	14.1 (10.0)	−0.9 (−2.6, 0.9)	18.0	−28, 25	147 (38)
Normal (BMI < 25.0 kg/m^2^)	15.0 (10.1)	Ref.	18.0	−25, 30	245 (62)
Leisure-time physical exercise					
At least 1×/week	15.5 (8.5)	−1.8 (−3.8, 0.2)	18.0	−18, 30	182 (46)
Never	13.7 (11.5)	Ref.	18.0	−28, 29	210 (54)
Smoking					
Smoker	13.2 (11.4)	−1.8 (−3.8, 0.1)	17.0	−28, 29	96 (24)
Non-smoker	15.1 (9.6)	Ref.	18.0	−25, 30	296 (76)
Alcohol consumption					
Weekly or more often	14.1 (11.4)	−0.6 (−3.2, 2.0)	18	−28, 25	46 (12)
Less often	14.7 (9.9)	Ref.	18	−25, 30	346 (88)
All	14.6 (10.0)		18.0	−28, 30	392

**Table 2 ijerph-20-02067-t002:** Regression of self-assessed cold threshold on personal characteristics and work-related factors. Regression coefficients indicate the difference (°C) in self-assessed threshold temperature in each class compared with the reference class. Percentiles arranged from warmest (P_17_) to coldest (P_83_).

Characteristics	Percentile	Regression coefficient (95% CI) (°C)		
		All Participants (N = 392)		Subsample (N = 284)
		Crude	Adjusted for Personal and Work-Related Factors ^1^	Also Adjusted for Worksite Physical Conditions ^2^
Sex (men vs. women)				
Ordinary linear regression		−2.8 (−4.4, −1.1)	−0.1 (−2.0, 1.8)	−0.4 (−2.9, 2.1)
Quantile regression	P_83_	−9.0 (−13.0, −5.0)	−5.0 (−9.1, −0.9)	−5.0 (−9.5, −0.5)
	P_67_	0.0 (−1.8, 1.8)	0.0 (−1.8, 1.8)	0.0 (−2.2, 2.2)
	P_50_	0.0 (−2.7, 2.7)	2.9 (1.1, 4.6)	1.8 (−0.4, 4.1)
	P_33_	0.0 (−0.1, 0.1)	5.0 (3.6, 6.4)	1.5 (−0.7, 3.7)
	P_17_	−1.0 (−2.5, 0.5)	1.6 (0.1, 3.2)	2.0 (−0.3, 4.3)
Age (change/10 yrs)				
Ordinary linear regression		−0.1 (−1.8, 1.5)	−0.4 (−1.2, 0.5)	0.5 (−0.4, 1.4)
Quantile regression	P_83_	−2.0 (−6.8, 2.8)	0.0 (−1.3, 1.3)	0.0 (−1.2, 1.2)
	P_67_	−3.0 (−5.6, −0.4)	0.0 (−0.6, 0.6)	0.0 (−0.8, 0.8)
	P_50_	0.0 (−2.7, 2.7)	−0.3 (−0.8, 0.2)	0.3 (−0.3, 0.9)
	P_33_	0.0 (−0.1, 0.1)	0.0 (−0.4, 0.4)	0.0 (−0.5, 0.5)
	P_17_	1.0 (−0.5, 2.5)	0.3 (−0.3, 0.9)	0.5 (−0.5, 1.4)
Job category (office work vs. other work)				
Ordinary linear regression		6.9 (5.2, 8.6)	6.8 (4.5, 9.1)	10.5 (6.7, 14.2)
Quantile regression	P_83_	10.0 (6.7, 13.3)	10.0 (7.0, 13.0)	15.0 (9.5, 20.6)
	P_67_	10.0 (8.1, 11.9)	8.0 (6.0, 10.0)	10.0 (6.7, 13.3)
	P_50_	5.0 (4.0, 6.0)	7.0 (5.4, 8.7)	10.3 (8.3, 12.3)
	P_33_	2.0 (−0.0, 4.0)	7.0 (5.2, 8.8)	9.5 (7.7, 11.2)
	P_17_	4.0 (2.9, 5.1)	3.8 (2.3, 5.3)	7.2 (4.2, 10.1)
Physical work strain (sedentary work vs. other work)				
Ordinary linear regression		4.3 (2.6, 6.1)	2.0 (0.1, 3.8)	1.5 (−0.8, 3.8)
Quantile regression	P_83_	7.0 (2.7, 11.3)	3.0 (0.4, 5.6)	0.0 (−3.0, 3.0)
	P_67_	6.0 (3.6, 8.4)	2.0 (0.3, 3.7)	0.0 (−1.9, 1.9)
	P_50_	5.0 (3.5, 6.5)	1.3 (−0.2, 2.8)	0.6 (−1.2, 2.3)
	P_33_	0.0 (−1.1, 1.1)	2.0 (0.6, 3.4)	1.5 (−0.1, 3.1)
	P_17_	2.0 (0.4, 3.6)	0.5 (−0.7, 1.9)	2.0 (0.1, 3.8)
Thermal insulation of clothing (clo ≤ 1.15 vs. clo > 1.15)				
Orinary linear regression		3.1 (2.1, 5.4)	−0.9 (−2.8, 1.0)	−1.6 (−3.6, 0.4)
Quantile regression	P_83_	4.0 (0.4, 7.6)	0.0 (−3.1, 3.1)	−4.0 (−8.2, 0.1)
	P_67_	3.0 (0.3, 5.7)	0.0 (−1.8, 1.8)	0.0 (−2.5, 2.5)
	P_50_	5.0 (3.5, 6.5)	−0.8 (−2.8, 1.2)	−1.0 (−2.7, 0.8)
	P_33_	0.0 (−1.4, 1.4)	−2.0 (−3.6, −0.4)	−1.5 (−3.1, 0.1)
	P_17_	3.0 (1.5, 4.5)	−0.2 (−1.6, 1.1)	−1.2 (−3.2. 0.9)
Alcohol consumption (weekly vs. less often)				
Ordinary linear regression		−0.6 (−3.2, 2.0)	1.7 (−0.9, 4.4)	−0.8 (−3.6, 1.9)
Quantile regression	P_83_	−5.0 (−13.0, 3.0)	4.0 (−2.8, 10.8)	1.0 (−6.1, 8.1)
	P_67_	3.0 (−0.5, 6.5)	5.0 (1.5, 8.5)	0.0 (−3.3, 3.3)
	P_50_	0.0 (−2.8, 2.8)	1.8 (−1.0, 4.6)	−0.8 (−3.0, 1.4)
	P_33_	0.0 (−0.5, 0.5)	0.0 (−1.3, 1.3)	−1.5 (−3.2, 0.0)
	P_17_	−2.0 (−4.0, 0.0)	−0.6 (−2.7, 1.5)	−1.0 (−3.0, 0.9)

^1^ Adjusted for sex, age, job category, physical work strain, thermal insulation of clothing, and alcohol consumption, excluding the variable of interest. ^2^ Worksite cold (<10 °C), relative humidity (%), and air velocity (m/s).

## Data Availability

The data are confidential and cannot be shared.

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
