# Peer review of "Self-Assessed Threshold Temperature for Cold among Poultry Industry Workers in Thailand"

_ijerph, 2023, doi:10.3390/ijerph20032067_

Round 1
Reviewer 1 Report
The topic is interesting, it certainly has relevance. The research plan seems good. The methodology is good, result are satisfying. My minor observations are mentioned below:
1. Introduction sections need to be incorporated with more scientific literature part done in this direction across worldwide.
2. If you provide a questionaries’ survey figure, that would be very nice. For example Fig2 in this paper: https://doi.org/10.1016/j.uclim.2021.100993
3. Detailed information of measuring instruments like their sensitivity etc. can be given in a tabular format.
4. The outer boundary from all the figures should be removed.
5. Axis of Figure 1 should be labelled.
6. If possible try to adjust the supplementary tables in the main manuscript.
7. Discussion section is nicely written.
8. Conclusion should be a separate part not within the discussion part.
Author Response
- Anwswers marked as bullet points
The topic is interesting, it certainly has relevance. The research plan seems good. The methodology is good, result are satisfying. My minor observations are mentioned below:
- Introduction sections need to be incorporated with more scientific literature part done in this direction across worldwide.
- Some more literature has been added to Introduction. However, we did not find any references specifically related to self-assessed cold temperature. Most studies in this field focus on how environmental cold is related to symptoms and complaints.
- If you provide a questionaries’ survey figure, that would be very nice. For example Fig2 in this paper: https://doi.org/10.1016/j.uclim.2021.100993
- Appendix Figure 1 now shows the questions used in this study.
- Detailed information of measuring instruments like their sensitivity etc. can be given in a tabular format.
- The technical details of the instruments are now in Appendix Table A1.
- The outer boundary from all the figures should be removed.
- The outer boundary has been removed from all the figures
- Axis of Figure 1 should be labelled.
- Both axes of Figure 1 have been re-labelled
- If possible try to adjust the supplementary tables in the main manuscript.’
- It proved quite difficult to incorporate all the supplementary to the text.
- Discussion section is nicely written.
- We appreciate this comment.
- Conclusion should be a separate part not within the discussion part.
- Conclusion is now a separate part (5. Conclusions)
Reviewer 2 Report
This is a fine paper, worth publishing provided minor presentation details are corrected in the bibliography. See comments in the attached report.

Author Response
- Answers marked as bullet points
This is a fine paper, worth publishing provided minor presentation details are corrected in the bibliography. See comments in the attached report.
(Attachment:) This is a good quality paper. It is well written, and all of its sections (introduction, methodology, discussion, etc.) are adequate. I am not familiar with the thermal insulation of clothing calculation formula, so I don’t know whether it is correct or not. Otherwise, the methodology as described seems adequate. Clearly this is one of a series or papers by the same authors on more or less the same. subject. Given that we haven’t necessarily read the previous ones, a bit more details would be appreciated about the population database used. As regards the format, the way the references are presented in the bibliography should be standardized; why are some authors’ names underlined?
I consider this paper can be published as it is.
- The formula to calculate the basic thermal insulation of clothing is given in ISO 9920 standard (reference 21) and was checked during the revision.
- There is a detailed description of the sample in reference [1] at https://pubmed.ncbi.nlm.nih.gov/32887559/, especially in Additional files 1 and 2. All details are freely available through the link (now given in the reference list), and therefore it may be unnecessary to add much more details to the body text. However, some more description of the work has been added to the study population paragraph, section 2.1.
- Some authors’ names were mistakenly underlined – now corrected
Reviewer 3 Report
The manuscript by Laohaudomchok et al., analysed the cold threshold self-assessed by the employees of poultry industry in Thailand. The study is location specific and highlight significant finding but needs the following clarifications:
1. The study is dependent on the human subject based on a questionnaire, however, the authors have not provided the full questionnaire. Its not clear weather the same questions were asked from all the subjects or influenced by the interviewer
2. While comparing the different groups in the study for example effect of obesity on cold threshold, the statistical significance of the observations are not clear in the manuscript
3. In conclusion the authors should mention the implications of the observed results
Author Response
- Answers are marked as bullet points
The manuscript by Laohaudomchok et al., analysed the cold threshold self-assessed by the employees of poultry industry in Thailand. The study is location specific and highlight significant finding but needs the following clarifications:
- The study is dependent on the human subject based on a questionnaire, however, the authors have not provided the full questionnaire. Its not clear weather the same questions were asked from all the subjects or influenced by the interviewer
- The questions used are now in Appendix Figure 1. All participants were asked the same questions.
- While comparing the different groups in the study for example effect of obesity on cold threshold, the statistical significance of the observations are not clear in the manuscript
- We are reluctant to report statistical significance or p values, rather we use confidence intervals. The use of p-values is discouraged ̶ see e.g. the special issue on the “problem of p-values” in the American Statistician, 2019 (https://www.tandfonline.com/toc/utas20/73/sup1). Briefly, the p-values (“statistical significance”) only indicate how likely it is that a result like this could have been generated by change. Confidence intervals give the precision of the estimate and thus contain more information than p-values alone. For example, among office workers the self-assessed cold temperature (CT) was 6.9 ºC higher than that among other workers although the difference could be as low as 5.2 ºC and could be as high as 8.6 ºC. In this case, the p-value was ~0.000, but it obviously gives less information than CI (5.2 ºC to 8.6 ºC). Among the obese individuals, CT was 0.9 ºC lower among the obese compared with the normal-weight subjects, but the estimate was rather inaccurate since CI was -2.6 to9. The p-value for BMI was ~0.540 but it is clearly less informative than CI.
- In conclusion the authors should mention the implications of the observed results
- We think that the conclusion section lists the implications of the results. The principal message is that individual assessments of what is cold temperature are meaningful with respect to adverse effects of low temperatures at workplace, some groups of workers being more sensitive than others, in different degrees depending on whether the self-assessed temperature is very cold, moderately cold or slightly cold. The conclusion sections points to sensitive groups of workers that need more protection: office staff, sedentary workers, male workers (even moderate cold), female workers (especially severe cold). These subgroups need intensified advice, since the adverse effect of cold, including the increased risk of actual disease events in a longer run, are well-known, and these effects are preventable by adequate clothing and other means listed in the conclusion section. We also state that research-based recommendations and standards for cold work which do not exist in Thailand should be established.
Reviewer 4 Report
Comments and suggestions
This manuscript aims to investigate the self-assessed threshold temperature for cold in the workplace. It conducted a survey of 392 chicken industry workers in Thailand that they regard as cold threshold and compared subgroups of workers. This study determines the self-assessed cold temperature with breakdowns according to sex, age, job category, education, physical work strain, thermal insulation of clothing, obesity, alcohol consumption, temperature, relative humidity, and air velocity among poultry industry workers. This manuscript has a well-structured description and an elaborated and comprehensive discussion. In practice, identifying sensitive groups of workers is useful because cold exposure adversely affects physical and mental performance and work capacity, and it increases the risk of cold injuries, accidents, and cardiac and respiratory events. This study also provided recommendations and pieces of advice in the end conclusions based on the results.
1. Introduction part: Plenty of previous references focused on Thai studies. The authors can put some references from other tropical countries' results in the introduction part. It can provide readers with different points of view regarding this topic.
2. Material and methods:
This part was addressed in very detail and abundant in this manuscript. Based on your address, there are many workers in northeastern Thailand, totaling 13,092 men and women. Why did this study adopt convenience sampling to collect data? Please add this important information to the revised manuscript.
Convenience samples are quite prone to research bias. Since the researcher draws the sample based on convenience and not equal probability, convenience samples never result in a statistically balanced selection of the population. This leads to sampling bias. We understand that the author has addressed the limitation of convenience sampling in the manuscript.
On page 3: (line 142)
The coefficients were calculated using the rq function available in R software, release 4.1.2 (https://CRAN.R- project.org/). This website can’t link, please confirm it.
3. Results
The data is clearly demonstrated in the manuscript.
4. Discussion
Did this study collect other demographic information, such as participants’ health status or medical history (acute or chronic diseases)? Moreover, female workers in their menstrual period may be more sensitive to cold. If this study has collected these variables, please add this important information in the revised manuscript.
Author Response
- Answers marked as bullet points
This manuscript aims to investigate the self-assessed threshold temperature for cold in the workplace. It conducted a survey of 392 chicken industry workers in Thailand that they regard as cold threshold and compared subgroups of workers. This study determines the self-assessed cold temperature with breakdowns according to sex, age, job category, education, physical work strain, thermal insulation of clothing, obesity, alcohol consumption, temperature, relative humidity, and air velocity among poultry industry workers. This manuscript has a well-structured description and an elaborated and comprehensive discussion. In practice, identifying sensitive groups of workers is useful because cold exposure adversely affects physical and mental performance and work capacity, and it increases the risk of cold injuries, accidents, and cardiac and respiratory events. This study also provided recommendations and pieces of advice in the end conclusions based on the results.
- Introduction part: Plenty of previous references focused on Thai studies. The authors can put some references from other tropical countries' results in the introduction part. It can provide readers with different points of view regarding this topic.
- We did not find any other references specifically related to self-assessed cold temperature. Most studies in this field focus on how environmental cold is related to symptoms and complaints, and one study has related reported cold symptoms to future disease events and deaths. However, some references on the adverse effects of cold have been added to Introduction.
- Material and methods:
This part was addressed in very detail and abundant in this manuscript. Based on your address, there are many workers in northeastern Thailand, totaling 13,092 men and women. Why did this study adopt convenience sampling to collect data? Please add this important information to the revised manuscript.
Convenience samples are quite prone to research bias. Since the researcher draws the sample based on convenience and not equal probability, convenience samples never result in a statistically balanced selection of the population. This leads to sampling bias. We understand that the author has addressed the limitation of convenience sampling in the manuscript.
- Drawing a probability sample proved too difficult due to the lack of reliable records (e.g. migrant workers). Even in case of a probability sample, significant numbers of dropouts and consequent selection would have been expected for reasons explained below.
- All participants in the factories concerned were offered an opportunity to participate. Guided by advance power calculations (see e.g. reference [1]), approximate quota were first determined for each factory. Before a worker could be recruited, a permission had to be obtained from the worker’s supervisor to interrupt the work, and another worker had to be searched to replace him/her. These factors would have affected the final sample also in case of a probability sampling. The final sample was also affected by the time schedules of the study team.
- As the referee notes, we recognize that the sampling scheme may have affected the results in an unknown way, but any major bias in groupwise differences is unlikely, because the results were adjusted for a reasonable number or potential confounding factors.
On page 3: (line 142)
The coefficients were calculated using the rq function available in R software, release 4.1.2 (https://CRAN.R- project.org/). This website can’t link, please confirm it.
- This works, we tested it: https://CRAN.R-project.org/
- You can also google “R software” and you will find several sites to access R.
- Results
The data is clearly demonstrated in the manuscript.
- Discussion
Did this study collect other demographic information, such as participants’ health status or medical history (acute or chronic diseases)? Moreover, female workers in their menstrual period may be more sensitive to cold. If this study has collected these variables, please add this important information in the revised manuscript.
- We asked physician-diagnosed medical conditions (now specified in Appendix Figure 1) and report the prevalence of major conditions (Personal characteristics subsection). The number of sick individuals was very small.
- The menstrual period was not asked, and it is now mentioned as a limitation. We recognize that in hot conditions (> 25°C)themenses onset may be affected by ambient temperature but this may not be true for cold conditions (Shoemaker JA, Refinetti R, Physiology & Behavior 1996;59(4/5):1001-1003).
Round 2
Reviewer 3 Report
The manuscript is in the acceptable format